# The Role of SVZ Stem Cells in Glioblastoma

**DOI:** 10.3390/cancers11040448

**Published:** 2019-03-29

**Authors:** Christine Altmann, Stefanie Keller, Mirko H. H. Schmidt

**Affiliations:** 1Institute for Microscopic Anatomy and Neurobiology, University Medical Center of the Johannes Gutenberg University, 55131 Mainz, Germany; Christine.Altmann@unimedizin-mainz.de (C.A.); Stefanie.Keller@unimedizin-mainz.de (S.K.); 2German Cancer Consortium (DKTK), partner site Frankfurt/Mainz, 60590 Frankfurt/55131 Mainz, Germany; 3German Cancer Research Center (DKFZ), 69120 Heidelberg, Germany

**Keywords:** glioblastoma, subventricular zone, neural stem cells, neurogenesis, brain tumor stem cells, therapy

## Abstract

As most common primary brain cancer, glioblastoma is also the most aggressive and malignant form of cancer in the adult central nervous system. Glioblastomas are genetic and transcriptional heterogeneous tumors, which in spite of intensive research are poorly understood. Over the years conventional therapies failed to affect a cure, resulting in low survival rates of affected patients. To improve the clinical outcome, an important approach is to identify the cells of origin. One potential source for these are neural stem cells (NSCs) located in the subventricular zone, which is one of two niches in the adult nervous system where NSCs with the capacity of self-renewal and proliferation reside. These cells normally give rise to neuronal as well as glial progenitor cells. This review summarizes current findings about links between NSCs and cancer stem cells in glioblastoma and discusses current therapeutic approaches, which arise as a result of identifying the cell of origin in glioblastoma.

## 1. Introduction

Glioblastoma (GB) is the most frequent form of brain tumor in adults and is associated with a poor prognosis and a short median patient survival [1]. Limited therapeutic options, combined with a poor response to currently used therapies, increased the pressure to discover new genetic, epigenetic and molecular pathways involved in GB to create new therapies. One of the most significant questions in GB research is aiming at the hierarchical organization and the cell of origin. Conventional theories state that cancer arises from an accumulation of somatic mutations, resulting in uncontrolled proliferation as well as selective growth advantage [2,3]. Most commonly, cancer occurs in epithelial tissues [4]. Whether a tumor originates from a differentiated cell, which regains the ability to proliferate, or whether it originates from a stem cell, which already has the capacity to proliferate, is not fully resolved, and depends on the tissue and the tumor itself. The existence of brain tumor propagating cells (BTPCs) and their molecular, genetic, and epigenetic footprint could open new ways of therapeutic approaches. In the last years, diverse tumors could be retraced to mutations in stem cells [4] and various studies have suggested that NSCs might be the cells of origin of GB, including mutated astrocyte-like NSCs from the SVZ [5,6,7,8]. Recent studies reported from clinics and mouse models that glioblastoma arise from migration of mutated astrocyte-like NSCs from the SVZ [8].

## 2. Glioblastoma

### 2.1. General Facts

Glioma is an umbrella term, compromising around 30 percent of all brain tumors that are thought to grow from intrinsic glia cells. As an umbrella term glioma consolidates different types of tumors including ependymoma, astrocytoma, and oligodendroglioma, which vary in their symptoms, aggressiveness, malignancy, and treatment strategy.

Glioblastoma multiforme (GB) belongs to the category of astrocytoma, is the most common and most aggressive of all malignant glial tumor in adults [1], and is less common in children [9]. Based on the World Health Organization classification, GB is the most malignant form of glioma and is classified as a grade IV tumor (ICD-O 9440/3) [10]. GB can be divided into primary (arising de novo) or secondary (developed from a pre-existing tumor) intrinsic brain tumor, however, 90% of all GB are primary [9]. Specific mutations in the gene of isocitrate dehydrogenase (IDH) 1/2 are characteristic for secondary glioblastomas, which are more frequent in younger patients. High invasiveness of GB is recorded, with tumor cells mainly spreading into distinct brain regions, whereas metastasis into other organs is infrequent [1].

Diagnosis of GB comes with a poor prognosis with high morbidity and mortality [1]. The median survival of patients diagnosed with GB and treated with the common medication is only 12 to 15 months [1]. GB can occur in each age group; however, most of the patients are between 45–75 years old. Gliomas are mainly located in the cerebral cortex of adult brains, with 40% in the frontal lobe, followed by the temporal lobe (29%), the parietal lobe (14%), the occipital lobe (3%) and 14 % of gliomas are positioned in deeper brain structures [11].

### 2.2. Genetic Alterations

GB features a complex pathogenesis that involves mutations and alterations of several key cellular pathways, associated with cell proliferation, angiogenesis, migration, and survival [9]. However, the lack of effective therapies increases the importance to understand pathogenesis in detail. Cellular signaling pathways involved in GB are reviewed in [9,12].

The most common mutations in GB are found in p53 (85.3–87%) [13,14], the epidermal growth factor receptor (EGFR) (45–57%) [13,14,15], the platelet-derived growth factor receptor (PDGFR) (60%) [16,17] the mouse double minute homolog 2 (10–15%) (MDM2) [18], the phosphatase and tensin homolog (PTEN) gene (20–34%) [19,20], the RTK/Ras/PI3K signaling pathway (86–89.6%) [13,14] and in the pRB signaling pathway (77–78.9%) [13,16].

### 2.3. Conventional Therapy

The current therapy of GB is limited and inefficient and focuses on surgical resection of as much of the tumorigenic tissue as possible with subsequent radiation- and chemotherapy, hereby mostly using oral alkylating agent temozolomide [21]. However, this therapy strategy is insufficient and no adequate cure for GB was described, yet. One major disadvantage of GB is the tissue-distribution pattern of the tumors resulting from dispersion of the tumors cells in the neighboring brain tissue [22,23]. This characteristic hinders complete surgical resection, increases recurrence rate and thereby reduces healing abilities.

Despite ongoing research, survival of GB patients could not be increased in the last years. However, there are rare case reports about long-term survival and zero recurrence of single patients with GB, which are explained by young age, as well as aggressive and complete surgical removal [24,25]. These cases, as rarely as they are, rise the hope that GB can be cured and show that more research effort and innovative treatment approaches are desperately needed to better understand the tumor development and progression.

## 3. Neural Stem Cell Niches in the Adult Brain

Neural stem cells (NSCs), a subpopulation of astroglial cells, are self-renewing cells with the capacity to differentiate into multiple neural cell types like neurons and glial cells (astrocytes and oligodendrocytes) (reviewed in [26]). During development, NSCs are obligatory for the formation of the nervous system. They are most active in this period; however, since 1992 it is described that NSCs can also be found in the adult brain. Here, small populations of NSCs are located in specific stem cell niches that divide occasionally to generate differentiated cells including neurons (neurogenesis) and glial cells (gliogenesis) [27,28].

### 3.1. Adult Neurogenesis

NSCs have a relative slow division rate and generate progenitor cells, which are able to differentiate into one of the three major cell lineages of the brain: neurons, astrocytes and oligodendrocytes [26,29,30]. The two neurogenic niches in the adult rodent brain, which contain NSCs and produce new neural cells, are the subventricular zone (SVZ) of the lateral ventricle and the dentate gyrus of the hippocampus [31,32]. While adult neurogenesis in the SVZ of humans is widely accepted [33,34], it is still under discussion in the hippocampus [33,35,36,37]. Recently, Sorrells et al. showed that neurogenesis in the dentate gyrus declines sharply in children and that in the adult brain they could not detect any new neurons [36]. However, shortly after this publication, Boldrini et al. published a manuscript stating that neurogenesis still persists throughout life [37]. Thus, it is still controversial if the potential to produce new neurons still exists in the adult brain; but many studies show that it is drastically decreased as compared to embryonic stages [5,33,38,39]. During the last decades adult neurogenesis was also described in other mammalian brain regions, including the hypothalamus, the cortex, the striatum, and the amygdala [40,41,42,43]. Furthermore, an increase of neurogenesis in the adult brain was reported after injury and in disease [44,45,46,47,48].

The impact and consequence of adult neurogenesis in humans is largely unknown and can only be estimated. However, many studies suggest that the generation of new neurons in the adult brain could be important for learning and memory, degradation as well as regeneration processes underlying aging, injury and diseases, including dementia and cancer [49,50,51,52,53,54].

### 3.2. Stem Cells in the Subventricular Zone

The SVZ is the largest neurogenic niche in the adult mammalian brain. It is located at the border of both lateral cerebral ventricles, fitting in between the ependyma and the parenchyma of the striatum. The NSCs in the SVZ are found in an astrocytic ribbon in the sub-ependymal zone. They are surrounded by ependymal cells, vascular endothelial cells, astrocytes, and oligodendrocytes [5], which are important to support the stem cells and to control the proliferation rate [55,56]. Next to the SVZ, the brain parenchyma is located, which is mostly composed of differentiated neurons and glia cells.

In adult rodents, the SVZ contains four major cell types: ependymal cells, NSCs, fast proliferation precursors and neuroblasts. The ependymal cells form a monolayer-border to the ventricle. This layer is followed by the other three cell types, which are not arranged in layers and keep close contact to the ependymal layer. Astrocyte-like NSCs have an apical cilium, which extends into the ventricle lumen and might influence cell proliferation and differentiation [57,58,59]. These astrocyte-like NSCs (type-B cells) occasionally give rise to multipotent intermediate progenitors (type-C cells), which correspond to transit-amplifying cells that further divide to generate neuroblasts (type-A cells) (reviewed in detail in [60]). In rodents, these neuroblasts migrate in chains along the rostral migratory stream (RMS) to the cortical layer of the olfactory bulb (OB) [61,62], where they differentiate into interneurons and integrate into the pre-existing neural circuitry [61,62,63,64].

NSCs in the adult human SVZ were first described in 2004 by Sanai et al. [5], followed by several groups, which identified and isolated human NSCs from the anterior SVZ (Figure 1) [65,66,67]. In the human SVZ, there are some differences in the anatomical organization and in the behavior of NSCs compared to rodents:

(I) The human SVZ features a hypocellular gap between the ependymal layer and the neurogenic astrocytes, which mainly contains ependymal and astrocytic expansions [5,38,68]. This layer is missing in rodents.

(II) The SVZ in humans can be separated into four distinct layers, whereas the rodent SVZ is not separated into layers. The first layer (ependymal layer, layer I) lines the ventricle and is composed of ependymal cells that possess several apical microvilli and basal expansions [38]. This layer is followed by the hypocellular gap forming the second layer (hypocellular layer, layer II) which is characterized by the absence of cell bodies and has an unknown function. As it mainly contains astrocytic and ependymal processes, one theory would be that these are forming connections in layer II [60,69,70]. The third layer (astrocytic ribbon, layer III) mainly contains the astrocyte-like NSCs and neuroblasts [5], whereas the forth layer, the transitional zone (layer IV) mainly contains myelinated axons and oligodendrocytes [38].

(III) In the human SVZ, multipotent transit-amplifying cells (in rodents: type-C cells) are missing. Only NSCs and neuroblasts can be found in the human SVZ.

(IV) The migration of neural progenitor cells from the SVZ along the RMS to the olfactory bulb is still under discussion and the migration of non-neuronal progenitor cells is at least reduced [5,33,34,67,71], whereas migration of these in rodents is verified. In the OB, mostly new oligodendrocytes from the SVZ are needed to maintain the myelin sheath of the neurons [72,73]. The fate of newborn neurons is unclear. It was shown by Ernst et al. in 2014 that in the adult human brain, new neurons from the SVZ are integrated not into the OB but in form of interneurons into the striatum [74]. Thus, the SVZ is still considered as an important pool of neuronal and glial progenitor cells in the adult mammalian brain, which provides an opportunity for neuroregenerative repair, learning, and memory [75,76,77,78,79]. However, this pool was also implicated in other conditions such as injury, neurodegeneration and cancer [80,81,82,83,84].

## 4. Cancer Stem Cells

Gradually, many studies have reported that tumor cells are not homogenous, but that a tumor consists of a variety of cell types forming a hierarchical organization containing slowly dividing cancer stem cells (CSCs) rapidly dividing precursor cells and non-dividing differentiated cells [85,86]. This cellular hierarchy in tumors was identified by genetic, molecular, epigenetic as well as behavioral variations of the cells. But what is the origin of these different tumor cells?

### 4.1. Cancer Stem Cell Theories

The transformation of a cell into a tumorigenic cell includes multiple mutations. There are two prominent theories about the origin of cancer cells. One property of at least some cancer cells is the ability to divide. However, do cancer cells acquire this ability or do they arise from a stem cell population that already possesses this ability?

The first theory about the origin of CSCs states that any body cell can become a cancer stem cell by mutation, meaning that already differentiated, somatic cells become tumorigenic. Therefore, an accumulation of mutations is needed in oncogenes (gain of function) or tumor suppressor genes (loss of function), which regulate cell growth, to transform somatic cells into CSCs [87,88,89]. These mutations occur through replication errors or DNA damage, combined with a missing or incorrect repair mechanism [90]. However, many critics on this conventional theory emerged in the last decades, because of the unlikelihood that various mutations occur in one mature (non-dividing) cell with a limited lifespan. These critics favor the second theory, which is called cancer stem cell theory. This theory is based on the self-renewal ability of stem cells or progenitor cells and states that CSCs arise through oncogenic mutation in stem cells. The remaining of this review will deal with this second cancer stem cell theory, including the influence of CSCs on glioblastoma, their origin in the SVZ as well as possible treatment therapies targeting CSCs.

The idea of stem cells derived CSCs was minted by studies using human leukemia cancer cells, which were transferred into immunodeficient mice [91,92]. When characterizing these cells, the authors found that the cells were quite heterogeneous and only a minor portion had the potential of producing leukemia in mice. This suggests that not all cancer cells but only the slowly dividing stem cells have the potential to reproduce the tumor itself [91,93]. Another study addressed breast cancer cells and described the heterogeneous phenotype of the cells. Only a limited number of cells in the tumor displayed tumorigenic potential which they identified by cell surface markers (CD44^+^ CD24^−/low^ lineage^−^) [94]. Thus, targeting these cells by cancer therapy would be most promising.

In addition to their tumorigenic properties and extensive proliferative potential, CSCs share various qualities with normal stem cells: (I) The capacity of multipotency, meaning the ability to differentiate into multiple lineages, self-renewal, and the capacity to divide into either new stem cells or into differentiated cells [86]. (II) A low self-renewal rate and rare occurrence (only one in a million cells) [92,93,95,96]. (III) A strict control by their microenvironment to regulate the balance between proliferation and cell death [97,98]. (IV) The usage of similar signaling pathways [85].

### 4.2. Neural Brain Tumor Propagating Cells

The hypothesis of CSCs can also be extended to brain tumors, here referred to as brain tumor propagating cells (BTPCs), however, with some minor deviations. As discussed above, stem cells are scarce in the adult brain and can only grow in protective stem cells niches, including the hippocampus and the SVZ [31,32]. These NSCs already possess the ability to proliferate and thus they could transform more easily and rapidly into BTPCs than any other post-mitotic neural cell in the brain [99]. After certain variations, neural precursor cells could become BTPCs (Figure 2). However, other than their offspring, NSCs normally do not leave their neurogenic niches. One hypothesis would be that BTPCs originate from a mutation or deregulation that enables the NSCs to migrate and leave the niche. This exit and a subsequent dysregulation of the stem cell might result in unpredictable proliferation and thus tumorigenesis [100,101,102,103].

Like other stem cells, NSCs have the ability of self-renewal and by asymmetric cell division, another NSC and differentiated daughter cell is created. To ensure that the number of NSCs remains constant, proliferation is strictly regulated by intrinsic and extrinsic factors from proliferative cells themselves, neighboring cells, as well as the adjacent blood vessels (reviewed in [104]). Disruption/mutation of these regulatory mechanisms could result in creation of a BTPC. However, brain tumors are not found in the SVZ itself, but rather in the cerebral cortex of adult brains. Thus, besides various tumorigenic mutations, the mutated neural precursor cells need a signal to migrate into other brain regions.

Tumorigenic mutations in migratory SVZ neural precursor cells would be of advantage, because of their ability to migrate over long distances [61,62]. However, in the adult human brain, migration of neuroblasts to the OB is limited, which raises the question if these cells still have the ability to migrate [33,34,67]. Brain injuries, like ischemia, increase the number of neural precursors and some NSCs migrate into the lesion site to increase regeneration, suggesting that the NSCs themselves are able to migrate [105,106]. Furthermore, BTPCs could migrate along tumor microtubes, which have been described as ultra-long membrane protrusions combining BTPC niches scattered in the brain [107,108]. Those BTPC niches are similar to NSC niches with a specific protective microenvironment composed of specific cell types. Based on their location within the tumor and their composition, they are mostly described as peri-vascular, necrotic and hypoxic niches [101,109]. BTPCs in peri-vascular niches are closely located to endothelial cells and express vascular endothelial growth factor VEGF, a well-known growth factor which regulates angiogenesis [110]. Those endothelial cells include capillaries, venules, lymph vessels or arterioles, whereby in case of arterioles, the niche can be directly specified as peri-arteriolar niche [111]. In hypoxic niches, the level of hypoxia-inducible factor HIF is increased, which can increase pro-angiogenic growth factors [112,113]. Most recently, a combined concept was presented, comprising all BTPC niches to one integral hypoxic peri-arteriolar niche model (reviewed in [114]).

Over the years, various studies on mouse models of brain tumors suggested that the cell of origin for brain tumors are NSCs located in the neurogenic niches [6,115,116,117,118]. These NSCs further proliferate in other brain regions, forming a tumor, which consist of BTPCs and more differentiated cancer cells. In the following the cancer stem cell theory is discussed in the context of glioblastoma.

## 5. The Cell of Origin in Glioblastoma

Over the years of glioblastoma research, various theories emerged about the cell of origin, including tissue-specific stem cells like NSCs or committed precursor cells, like astrocyte precursor cells (APCs) and oligodendrocyte precursor cells (OPCs) [119,120]. 

### 5.1. NSCs as the Cells of Origin in Glioblastoma

Gradually, several groups identified that glioblastomas are organized hierarchically and contain tumorigenic cells, which could correspond to BTPCs [121,122,123,124,125] with the basic BTPC and NSC characteristics like multi-lineage potency, self-renewal capacity and the ability of tumor initiation and migration [122,123,126,127,128]. Another evidence for an involvement of undifferentiated BTPCs in GB was provided by discovering that the cells share molecular pathways with NSCs, e.g., the Notch receptor activation. Notch 1 along with its signaling partners is important for NSC maintenance [129] and glioma cell survival [130]. Furthermore, BTPCs and NSCs both express characteristic genes, which can be found in human tumor tissue [125], including Nestin [131,132,133,134,135,136], CD133 [133,137], Sox [132], Musashi1 [136,138], Olig1/2 [139,140], Ras [141], Akt [141], and GFAP [142,143]. However, these genes are not exclusively expressed in NSCs and BTPCs, but can be found in various other cell types, e.g., GFAP is expressed in astrocytes and radial glial cells. Also, the inactivation of certain tumor suppression genes is similar, e.g., PTEN, which increases proliferation of NSCs as well as of tumor cells [144].

One of the neurogenic niches, the SVZ, came into the focus of BTPC research. Already in the early 21st century, GB was linked to the NSCs of the SVZ. Ignatova et al. isolated cancer cells from human glioblastoma, took them in culture and discovered that these tumors contain neurosphere-forming cells [121]. Those BTPC niches Comparing the structural organization of these tumor-derived spheres to those derived from adult human SVZ highlighted a similar hierarchy, concerning the general organization and an outward gradient of differentiation [145]. This gradient in neurospheres is believed to be important for maintaining the stem cell pool. However, for a long time, data from mouse models could not provide a direct link between SVZ NSCs and GB BTPCs.

One novel factor involved in both GB and SVZ NSCs is the extracellular matrix protein epidermal growth factor-like protein 7 (EGFL7), which was previously described to be secreted by endothelial cells as well as NSCs in the SVZ [146,147,148,149,150]. Loss of EGFL7 increased proliferation of NSCs and decreased cell differentiation in the SVZ, mainly via Notch signaling [146,148]. Additionally, the interaction of Notch 1 and EGFR increased survival of GB cells [147] via promoting angiogenesis [15]. The influence of EGFL7 in angiogenesis [149] also plays an important role in GB [150]. EGFL7 is secreted by glioma blood vessels and increases angiogenesis in GB, suggesting inhibition of EGFL7 as a possible treatment strategy [150].

Analyses of the survival of patients with GB revealed that it is crucial if the tumor has a direct connection to the SVZ [151,152]. Patients bearing tumors which were in contact to the SVZ had a shortened survival period [151,153,154,155], but the size of these tumors was not altered [156] and thereby did not influence the survival time [152]. Potentially, a contact to the SVZ might provide the tumor with a pool of tumorigenic stem cells, therefore increasing the growth and invasiveness of the tumor. Proteomic analyses by Gollapalli et al. revealed significant alterations in acute phase proteins (e.g., proteins-hemopexin, alpha-1-antichymotrypsin), lipid carrying proteins (e.g., apolipoprotein A1), cytoskeletal proteins (e.g., brain acid soluble protein 1, thymosin beta 4), lipid binding proteins, chaperones, and regulating proteins between tumors with SVZ contact and without [157]. However, gene expression studies showed that there is no evidence that tumors with contact to the SVZ are more likely to be stem-cell derived than tumors without any contact to the SVZ [153]. Following this, two scenarios were suggested: [I] Tumors without contact are not derived from SVZ NSCs or [II] NSCs from the SVZ migrate into other brain regions and only then start to form a tumor. Various studies are pointing to the second scenario [8,158], although it could not be explained why the survival of the patients is altered.

The origin of the tumor can influence the malignance and survival of the patients. A study in mice showed the induction of GB originating in NSCs of the SVZ increases tumor development and resistance to cancer drugs, giving a hind to the aggressiveness and the malignance of glioblastoma [159]. Furthermore, chronic inflammation in the SVZ could facilitate and accelerate mutations in NSCs and thus contribute to the transformation into BTPCs (reviewed in [99]). The SVZ represents a unique inflammatory niche, which differs significantly from other brain regions, due to close contact to the cerebrospinal fluid (CSF).

Recently, Lee et al. showed direct genetic evidence that brain BTPCs arise from cells of the SVZ using human patients and glioblastoma mouse models [8]. The authors performed single-cell sequencing and laser microdissection on tumor free SVZ tissue and tumor tissue and found a clonal relationship of driver mutations between the SVZ and GB-derived tissue. This is the first proof that SVZ NSCs indeed are the cell of origin in glioblastoma. Furthermore, the authors could show that astrocyte-like NSCs leave the SVZ and migrate into distant brain regions to form gliomas. Restoration of homeobox protein EMX2 expression, which is normally decreased in cancer tissue, but is also expressed in NSCs of the SVZ, affects the cell cycle in glioblastoma, leading to a cell cycle arrest and cell death [160]. A recent study of the tumorigenic potential in different stages of neuronal progenitor cells showed that the tumorigenic potential decreases with an increasing lineage restriction, hinting that mutations likely occur in early stages of neurogenesis, e.g., in NSCs [161]. Taken together, these latest reports on the involvement of SVZ derived NSCs in GB are a huge step to discover new therapeutic targets for the treatment of GB patients.

### 5.2. Committed Precursor Cells as the Cell of Origin in Glioblastoma

Another theory about the cell of origin states that glioblastoma BTPCs arise from committed precursor cells, like APCs or OPCs. Proliferating multipotent NSCs can create committed neuronal precursor cells that may further divide and differentiate into mature neurons and committed glial precursor cells (GPCs) that can further differentiate into OPCs to generate mature oligodendrocytes and into APCs to generate mature astrocytes [162,163].

For a long time, astrocytes were believed to be the only proliferating cells in the adult brain [164] und thus, were subject in glioblastoma research, also because of high GFAP levels in glioma tissue [165]. However, these mature astrocytes would need to dedifferentiate to become tumorigenic, which is possible but unlikely [166,167]. Furthermore, GFAP is also expressed by radial glial cells [168] and NSCs of the adult SVZ [55].

Lindberg et al. and Hide et al. introduced OPCs as the cell of origin by specific mouse models to study OPCs in glioblastoma development, like the MADM-based lineage tracing model to mutate sporadically p53/Nf1 [120,169]. Hide et al. suggested that transformation of both OPCs and NSCs could lead to formation of BTPCs with tumorigenic properties [120]. Additionally, Liu et al. noted that it is important to analyze premalignant stages of tumors to identify the cell of origin, because the tumor cell could acquire plasticity and veil their origin [119]. Further, they demonstrated that OPCs, but not NSC or any other NSC-derived lineage, show aberrant growth prior to malignancy. In OPCs are some overlapping marker expressions, like PDGFRα and NG2, which are involved in development of OPCs and are altered in glioma [13,170,171]. OPCs might form a stem cell niche at the tumor border, increasing chemo-radioresistance and promoting recurrence [171].

Taken together, the current data suggests that BTPCs might develop from various stem or progenitor cells, which needs to be considered when developing treatment strategies. NSCs, ASCs, OPCs, and GPCs might all be the cell of origin, which lead to the development of GB. However, most recent studies using state of the art techniques clearly point to an involvement of NSCs in GB [8].

## 6. Brain Tumor Propagating Cells as Target for Glioblastoma Treatment

Traditional glioma treatments include chemotherapy, radiation, and surgery. However, these approaches are limited due to multiple factors like the development of therapy resistance coming with a decreasing sensitivity of tumor cells towards ionizing radiation, the unspecific targeting of both cancer and healthy brain cells, as well as the recurrence of the tumor after treatment. Recent advances suggest that the failure of therapies might be caused by the assumption that the tumor consists of a homogenous group of cells. In fact, the tumor is distinguishable into many subgroups of cells varying on genetics, epigenetics and phenotype as well as their capabilities to replicate and migrate. Thus, elimination of BTPCs is a promising alternative to specifically target the cell of origin of the tumor, which raises the possibility to prevent tumor proliferation, tumor cell differentiation and BTPC regeneration.

### 6.1. Complications in Glioblastoma Therapy

Due to specific BTPC characteristics, like slow cell division rate, self-renewal properties, high capacity for DNA repairing and high expression of drug transporters, the identification and targeting of this cell population represents a challenge to this day. Moreover, BTPCs are capable of developing resistance mechanisms in multiple ways complicating conventional drug efficacies. High expression of ATP-binding cassette drug transporters can impede cytotoxic agents to enter the cell [172], resulting in resistance to different chemotherapeutic drugs including the commonly used alkylating agent temozolomide [173,174] and increasing the risk of tumor recurrence after the treatment [175]. Besides the chemo-resistance, BTPCs are capable of developing a radio-resistance by an increase in the activation of the DNA repair machinery, which is promoted by the expression of stem cell marker CD133 [176]. This combined chemo- and radio-resistance hampers a successful treatment and therefore many patients require combinational therapeutic strategies to improve the survival.

Another way how BTPCs escape especially surgery is by forming stem cell niches and using ultra-long membrane protrusions, tumor microtubes, which can be found in various brain tumors and can be used as migration routes for cells located in BTPCs niches scattered in the brain. Disconnecting these tumor microtubes was suggested as a new target for treatment of GB [107,108].

The brain and especially brain tumors are always considered as extremely difficult for treatment, due to the blood–brain barrier (BBB). The BBB normally hinders harmful substances and toxins to enter the brain via different cellular and molecular components as well as divers transport systems. However, the location of the SVZ at the border to the lateral ventricle introduces a new aspect to the system, the CSF, which is secreted by the choroid plexus, forming the blood–cerebrospinal fluid barrier (CSFB). This barrier is functionally distinct and is not as tight as the BBB; mostly all non-cellular substances can enter the CSF [177]. It raises an opportunity for future drugs to enter the brain via the CSF and thereby increase the range of possible treatment molecules. This new approach was already successfully applied to medulloblastoma, another brain tumor type [178].

### 6.2. BTPCs as Target in Glioblstoma

Over the years, different approaches to target BTPCs were developed and tested (Figure 3), mostly in combination with traditional treatments and with more or less successful outcome [179,180]. There are three major therapeutic strategies for targeting BTPCs: (I) Targeting specific cell surface markers, signaling pathways or BTPCs microenvironment, (II) the induction of apoptosis or stem cell differentiation, as well as inhibition of autophagy and (III) the application of vaccines or epigenetic drugs.

#### 6.2.1. Targeting Specific Cell Surface Markers of Signaling Pathways or the Microenvironment of BTPCs

The most prominent marker to target the cell surface with antibody-drug conjugates is CD133, which is expressed not only in brain tumors [124], but also in other cancer types like lung cancer [181] or colorectal cancer [182]. The expression of CD133, alone or together with proliferation marker Ki67, is correlated to a glioma patients’ prognosis with poor clinical outcome [137,183]. However, only recent studies found a population of BTPCs that does not express CD133 and displays a lower proliferation rate compared to the CD133-positive cell population, with equal tumorigenic characteristics [184]. Thus, targeting CD133-positive cells alone could spare some proliferating BTPCs and thereby increase the reoccurrence rate, but targeting CD133-positive cells could be combined with other treatment strategies.

Targeting aberrant signaling pathways is a common approach in all forms of cancer. As for many cancer cell types, Notch signaling, the Wnt/β-catenin pathway and the PI3K/Akt cascade are among the pathways, which are significantly up-regulated and in the focus for targeting BTPCs in glioblastoma [185,186,187]. As an example, the upregulation of the Notch pathway in glioma plays a significant role in increasing tumorigenesis by promoting proliferation and cell survival. Evidence emerged that it also directly regulates maintenance and cell differentiation of BTPCs [130,188]. Application of γ-secretase inhibitors to impair Notch signaling led to decreased growth of glioma, but also induced both neuronal and astrocytic differentiation in the stem cell niches and thereby decreased the number of CD133-positive BTPCs [189,190].

Another way to target BTPCs is to inhibit or modify the signaling from their microenvironment that sustains the growth of the cells and is necessary for fundamental processes e.g., the regulation of hypoxia and angiogenesis. These processes are closely linked, with angiogenesis being regulated through hypoxia. The hypoxia-inducible factor (HIF) is regulating numerous genes, including vascular endothelial growth factor (VEGF) expression. The overexpression of HIF-1α and HIF-2α have been linked to poor prognosis in many cancer types [191]. HIF-2α-positive cells are in close contact to BTPCs, which indicates that BTPCs might be able to self-regulate their maintenance through communication with surrounding cells and support of angiogenesis [113]. Different agents to target HIF proteins are in development with some being tested in clinical trials [191] and it was also previously shown that bevacizumab, a VEGF neutralizing antibody, could specifically inhibit the proangiogenic effects of BTPCs [192]. A combinational therapy, targeting both hypoxic cancer cells and angiogenesis, may therefore represent a promising strategy in depleting BTPCs.

#### 6.2.2. The Induction of Apoptosis, Autophagy or Stem Cell Differentiation

A further approach to diminish the number of BTPCs and to erase the tumors origin is the induction of apoptosis. Apoptosis includes a complex signaling network and the evasion of this system is crucial for the stem cell survival as well as tumor development. Agents to manipulate apoptotic pathways include the inhibition of tumor necrosis factor related apoptosis inducing ligand (TRAIL) and bortezomib, a proteasome inhibitor, which, given in combination, could trigger stem cell death in glioblastoma [193]. Also, both EGFR inhibitors, AG1478 and gefitinib, did not only decrease cell proliferation, but also induced apoptosis of BTPCs [194].

Autophagy, a process of degradation and recycling of cellular components, has also been studied in glioblastoma BTPCs, especially in the context of drug-therapy resistance [195]. Modulation of the autophagic process provides an opportunity to increase cell death and to interfere with the cell cycle in BTPCs. Impairment of the autophagic flux decreases the cell self-renewal capacity of BTPCs. Furthermore, it was reported that chemotherapy increases autophagy in cancer cells [196] and that combining cytotoxic drugs and autophagy inhibitors (e.g., chloroquine or quinacrine) increase sensitivity of BTPCs [197]. Combination of bevacizumab or temozolomide with autophagy inhibitor chloroquine increased efficiency of the chemotherapy and affected survival of BTPCs [198,199]. Another novel and promising approach is the autophagy inhibitor quinacrine, which is able to cross the blood-brain barrier and increases the responsiveness of BTPCs to temozolomide and thereby death of BTPCs [200].

Stem cell differentiation also displays a promising strategy to limit the number of cells with tumorigenic potential and to produce cells with higher sensitivity to radio- and chemotherapy. Among the anti-cancer drugs affecting cell differentiation are bone morphogenetic proteins (BMP), retinoic acid and histone deacetylase (HDAC) inhibitors [201]. The application of BMP has been shown to block glioma development and to reduce CD133-positive cells, probably by manipulating symmetric cell cycles that generate new stem cells [202].

#### 6.2.3. The Application of Vaccines or Epigenetic Drugs

HDAC inhibitors are belonging to the category of epigenetic drugs and target reversible histone acetylation and methylation. HDAC inhibitor vorinostat is approved by the Food and Drug Administration [203] and is blocking HDAC SirT1 (silencing information regulator), which both increased apoptosis and cell differentiation of CD133-positive glioma cells [204].

Immune cell therapy has a lot of potential and receives increasing attention in the field of cancer treatments. The dendritic cell vaccine ICT-107 consists of six synthetic peptides derived from tumor-associated and on glioma BTPCs overexpressed antigens (AIM-2, MAGE1, TRP-2, gp100, HER2/neu, and IL-13Rα2). It is currently tested in phase III studies and it was already reported that ICT-107 treatment decreased CD133-positive cells and prolonged survival of the patients [205].

#### 6.2.4. Radiation of the SVZ

Mutated tumor cells might reside in the SVZ and other BTPCs niches, forming a reservoir of BTPCs leading to increased tumor recurrence and decreased outcome of GB patients. An approach to overcome distribution of tumor cells as well as radio resistance would be to decrease the number of BTPCs in the SVZ by radiation. Studies in this field however report conflicting results, depending on the dose of radiation and the size of the resected area [128,206,207,208,209,210,211,212,213,214,215].

Besides controversy of the benefit of this treatment, side effects are difficult to estimate. Delivering high doses of radiation to healthy SVZ tissue could decrease quality of life by loss of memory. One recent study on a large group of patients could not find any improvement in survival [216]. This might indicate that BTPCs either do not reside in the SVZ or that there are other reservoirs of BTPCs (BTPC niches) outside the SVZ which lead to a regrow of the tumors.

All these therapeutical approaches are only a small selection of the broad range of agents under development and examination to target BTPCs in GB and how this, especially in combination with classical therapies, may bring benefits in erasing cancer stem cells for a successful treatment of GB patients.

## 7. Conclusions

Each year, 240,000 cases of glioma are recorded worldwide, with GB accounting for most of these cases. As the most aggressive and malignant form of primary brain cancer, GB leads to thousands of deaths per year. The current standard care of therapy consists of a combination of surgery, radiation, and chemotherapy but the therapeutical success remains poor. This may be due to invasive tumor cells infiltrating neighboring brain regions, intra-tumoral heterogeneity and the capacity of distinct tumor cell populations to develop therapy resistance mechanisms. In this review we summarized various studies, which show that major reasons for the almost inevitable tumor recurrence are BTPCs, which are often spared by conventional therapy. It is suggested that these cells originate from astrocytic-like NSCs in the SVZ with the ability to leave the niche and to migrate over long distances. Through division into either differentiated cells or new BTPCs, they can provide an infinite pool of tumor cells and, additionally, the cells’ chemo- and radio-resistance in turn leads to resistance of recurrent GB to standard therapy. The identification of BTPCs and their origin raises hope to identify new molecular, epigenetic and genetic characteristics for the development of combination therapies to erase tumor cells as well as BTPCs. Current research focuses on multiple ways to target BTPCs including targeting specific cell surface markers, signaling pathways or the microenvironment, inducing either apoptosis or cell differentiation, inhibiting autophagy or applying vaccines or epigenetic drugs. Some of the agents are already investigated in clinical trials, showing promising outcomes. Further research is still needed to verify the latest results and technical advantages will help to discover specific GB BTPC treatment to terminate tumor stem cell proliferation and thereby increase the survival of GB patients. Taken together, this leads towards the development of a cure for this dreadful disease.

## Figures and Tables

**Figure 1 cancers-11-00448-f001:**
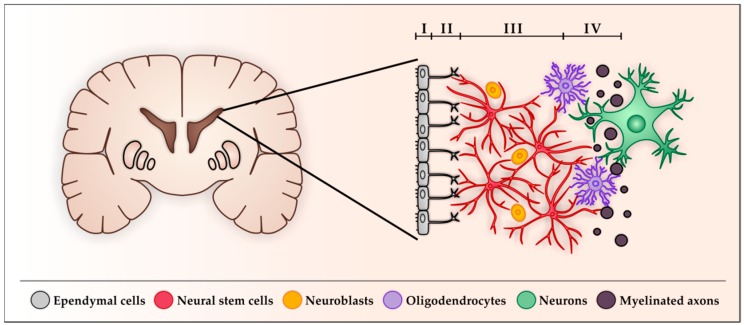
Schematic representation of the subventricular zone (SVZ) in a coronal view of the human brain and the adult human neurogenic niche in the sub-ependymal zone showing the cellular composition. The neurogenic niche in the SVZ can be divided into four layers (layer I–IV). The ependymal layer (layer I) separates the SVZ form the lumen of the ventricle by a thin single layer (monolayer) of ependymal cells (grey), which feature several apical microvilli and basal expansions. The ependyma is followed by the hypocellular layer (layer II), mainly containing astrocytic and ependymal processes. The astrocytic ribbon (layer III) primary contains astrocyte-like neural stem cells (NSCs) (red) and neuroblasts (orange), closely followed by the transitional zone (layer IV) which contains myelinated axons (black) and oligodendrocytes (violet). Thereafter the parenchyma begins, which mostly consists of neurons (green) and glia.

**Figure 2 cancers-11-00448-f002:**
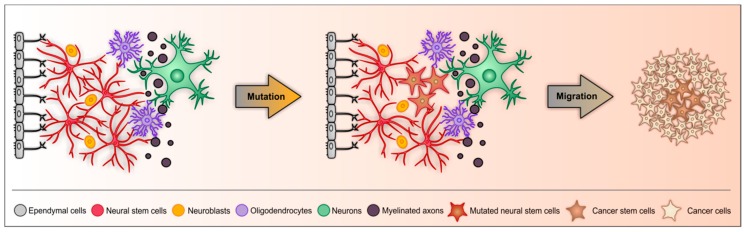
Schematic representation of the origin of brain tumor propagating cells (BTPCs) based on the cancer stem cell theory. In the SVZ of the adult brain, NSCs can be found in the sub-ependymal zone. There are two properties that NSCs need to achieve to become BTPCs. First, the cells need to become tumorigenic, including several mutations, resulting in mutated NSCs. Second these mutated NSCs have to be able to migrate over a long distance into other brain regions, where they form a tumor, which consists of BTPCs and more differentiated cancer cells which originate from the BTPCs themselves.

**Figure 3 cancers-11-00448-f003:**
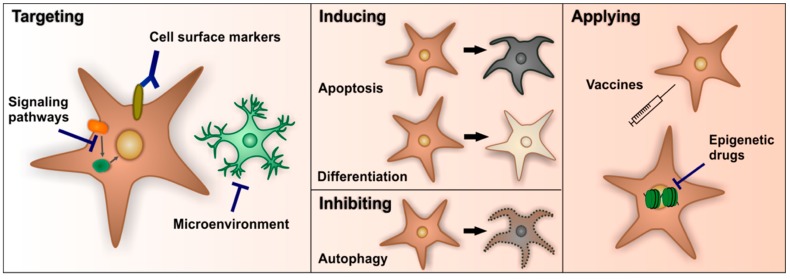
There are three therapeutic strategies to target brain tumor propagating cells (BTPCs). (I) Most common approaches include targeting specific cell surface markers, aberrantly up- or down-regulated signaling pathways or the BTPCs’ microenvironment. (II) The induction of apoptosis or stem cell differentiation as well as the inhibition of autophagy, which aims to reduce the number of BTPCs. (III) Application of vaccines or epigenetic drugs, which are still in development.

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
