# Peer review of "The Role of SVZ Stem Cells in Glioblastoma"

_cancers, 2019, doi:10.3390/cancers11040448_

Round 1

Reviewer 1 Report

A comprehensive review of the role of SVZ stem cells in glioblastoma was written by the authors. Background information and novel achievement of the development of glioblastoma were fairly mentioned in this manuscript. Only minor suggestion was raised to the authors for improvement.

1.     Accumulating evidence highlights the importance of SVZ stem cells in the development of glioblastoma. However, mutation of cancer cells transforming to stem cells might be also of possibility. A brief mention and citation of such hypothesis is recommended.

2.     6.2.2. The induction of apoptosis or stem cell differentiation is approach to diminish the number of BTPCs. Autophagy is also one way to diminish BTPC pool. Relevant description is helpful.

Author Response

Preamble

We thank the reviewer for the comments and the opportunity to significantly improve our manuscript. In this revision, we have addressed the concerns of the reviewer thoroughly. All changes in the text are indicated in red in the revised manuscript and also depicted in this point to point response. Further, the manuscript was improved by further proofreading.  Please find below our responses to the reviewer’s comments.

Reviewer #1:

A comprehensive review of the role of SVZ stem cells in glioblastoma was written by the authors. Background information and novel achievement of the development of glioblastoma were fairly mentioned in this manuscript. Only minor suggestion was raised to the authors for improvement.

1.      Accumulating evidence highlights the importance of SVZ stem cells in the development of glioblastoma. However, mutation of cancer cells transforming to stem cells might be also of possibility. A brief mention and citation of such hypothesis is recommended.

We thank the reviewer for the comment. Based on your suggestions we incorporated this thesis in more detail in the manuscript. We included a section in paragraph 4.1 cancer stem cell theories, reading:

“The first theory about the origin of CSCs states that any body cell can become a cancer stem cell by mutation, meaning that already differentiated, somatic cells become tumorigenic. Therefore, an accumulation of mutations is needed in oncogenes (gain of function) or tumor suppressor genes (loss of function), which regulate cell growth, to transform somatic cells into CSCs (Alt, Kellems et al. 1978, Rodenhuis 1992, Croce 2008). These mutations occur through replication errors or DNA damage, combined with a missing or incorrect repair mechanism (Martincorena and Campbell 2015).”

2.      6.2.2. The induction of apoptosis or stem cell differentiation is approach to diminish the number of BTPCs. Autophagy is also one way to diminish BTPC pool. Relevant description is helpful.

Including autophagy to possible therapeutic strategies was a great suggestion. We included this point in figure 3 and in paragraph 6.2.2, reading:

“Autophagy, a process of degradation and recycling of cellular components, has also been studied in glioblastoma BTPCs, especially in the context of drug-therapy resistance (Nazio, Bordi et al. 2019). Modulation of the autophagic process provides an opportunity to increase cell death and to interfere with the cell cycle in BTPCs. Impairment of the autophagic flux decreases the cell self-renewal capacity of BTPCs. Furthermore, it was reported that chemotherapy increases autophagy in cancer cells (Sui, Chen et al. 2013) and that combining cytotoxic drugs and autophagy inhibitors (e.g. chloroquine or quinacrine) increase sensitivity of BTPCs (Sun, Shen et al. 2016). Combination of bevacizumab or temozolomide with autophagy inhibitor chloroquine increased efficiency of the chemotherapy and affected survival of BTPCs (Encouse, Hee-Yeon et al. 2014, Huang, Song et al. 2017). Another novel and promising approach is the autophagy inhibitor quinacrine, which is able to cross the blood-brain barrier and increases the responsiveness of BTPCs to temozolomide and thereby death of BTPCs (Buccarelli, Marconi et al. 2018).”

Alt, F. W., R. E. Kellems, J. R. Bertino and R. T. Schimke (1978). "Selective multiplication of dihydrofolate reductase genes in methotrexate-resistant variants of cultured murine cells." J Biol Chem 253(5): 1357-1370.

Buccarelli, M., M. Marconi, S. Pacioni, I. De Pascalis, Q. G. D’Alessandris, M. Martini, B. Ascione, W. Malorni, L. M. Larocca, R. Pallini, L. Ricci-Vitiani and P. Matarrese (2018). "Inhibition of autophagy increases susceptibility of glioblastoma stem cells to temozolomide by igniting ferroptosis." Cell Death & Disease 9(8): 841.

Croce, C. M. (2008). "Oncogenes and Cancer." New England Journal of Medicine 358(5): 502-511.

Encouse, B. G., C. Hee-Yeon, J. Ardeshir, M. H. Florence, G. L. Stan, H. S. Axel and C. C. Thomas (2014). "Chloroquine enhances temozolomide cytotoxicity in malignant gliomas by blocking autophagy." Neurosurgical Focus FOC 37(6): E12.

Huang, H., J. Song, Z. Liu, L. Pan and G. Xu (2017). Autophagy activation promotes bevacizumab resistance in glioblastoma by suppressing Akt/mTOR signaling pathway.

Martincorena, I. and P. J. Campbell (2015). "Somatic mutation in cancer and normal cells." Science 349(6255): 1483-1489.

Nazio, F., M. Bordi, V. Cianfanelli, F. Locatelli and F. Cecconi (2019). "Autophagy and cancer stem cells: molecular mechanisms and therapeutic applications." Cell Death & Differentiation 26(4): 690-702.

Rodenhuis, S. (1992). "ras and human tumors." Semin Cancer Biol 3(4): 241-247.

Sui, X., R. Chen, Z. Wang, Z. Huang, N. Kong, M. Zhang, W. Han, F. Lou, J. Yang, Q. Zhang, X. Wang, C. He and H. Pan (2013). "Autophagy and chemotherapy resistance: a promising therapeutic target for cancer treatment." Cell Death &Amp; Disease 4: e838.

Sun, R., S. Shen, Y.-J. Zhang, C.-F. Xu, Z.-T. Cao, L.-P. Wen and J. Wang (2016). "Nanoparticle-facilitated autophagy inhibition promotes the efficacy of chemotherapeutics against breast cancer stem cells." Biomaterials 103: 44-55.

Reviewer 2 Report

The review of Altmann, Keller and Schmidt entitled “The role of SVZ stem cells in glioblastoma” aimed to summarize data which support the hypothesis that glioblastoma (GBM) cells derive from neural stem cells (NSC) present in the subventricular zone (SVZ). 

In this review, the authors stated that several data strongly suggest that NSC could be the cell origin of GBM.   

After remembering the clinical situation of GBM patients with their rather limited life expectancy when diagnosis is performed, authors rapidly describe the molecular pathology of GBM and the actual conventional therapy. Then, they move to the NSC and the adult neurogenesis, its possible regulation in various common neurological diseases and its physiological role.  However, although they cite the paper, they do not evoke the possibility that in human brain, this adult neurogenesis is questioned (Sorrells et al., Nature, 2018, 555, 377-381). Indeed, they put this citation in a sentence related to the fact that dentatus gyrus as a neurogenic zone is a still a matter of debate (line 102-104). 

Then authors carefully described the adult SVZ in human and listed the differences known to exist between human and rodent brains. Then the authors move to the description of Brain Tumor Propagating Cells (BTPC) also called GBM stem cells by others. Authors decided to focus their on “cancer stem cells theory” stipulating that mutations arouse in stem cells or in progenitor cells are able to explain the cancer onset.  

Then authors describe general features commonly accepted for Tumor stem cells.  They support the theory that BPTCs derive from NSCs and the mutations acquired allow them to leave the niche and continue to proliferate outside of it. Then the authors discussed the ability of BTPC to migrate a long distance as many GBM appears in various brain regions, away from the SVZ. 

Then, they compare at the molecular level genes express by both NSC and BPTC.  This is interesting but authors forget to mention that all these genes and/or markers are also expressed by other cell types in the brain (see major point 1). 

Authors also bring importance to the fact that both NSC and BTPC could be cultivated as spheres.  This characteristic is share by many progenitors or stem cells cell types and personally, I don’t agree with such an argument to claim that BPTC are derived from NSC (see point 2). 

Then authors move on an interesting section about the BTPC as a new therapeutic target. 

This section is interesting but is far from the main subject of the review.  Indeed, the only link of SVZ here is the fact that given the brain blood barrier, maybe the blood-cerebro spinal fluid barrier could be considered as an interesting way of entry for a drug. 

Finally, authors discussed the possible irradiation targeting the SVZ by only saying that there are controversial results which have been published in this field of clinical research. They do not evoke the radio-resistance of BTPC nor the possible role radio-protective role of SVZ.

In conclusion, this review could be interesting but rapidly, several oversimplifications made by authors decrease the potential utility of such a review.

Major points:

1)    In Line 241 to 245, authors compare the genes expressed by both NSC and BTPC. They should also mention that these markers could also be expressed by other cell types.  In others words, these are not bona fidemarkers of NSXC and/or BTPC.

2)    The sphere forming capacity is a functional marker of immature cells, normal and cancerous.  The paragraph between line 254 to 265 should be more carefully discussed. Indeed, to sustain the affirmation that glioma arise from NSC based on this property and on a paper published in 2003 is rather weak (line 259-260).

3)    The first main criticism to this review is the fact that authors do not consider other niches than VZ for glioblastoma stem cells, like vascular or the hypoxic peri-arterioalr niche (several references).  Indeed, it would have been useful considering BTPC and SVZ to compare those glioblastoma-specific niche to SVZ or the normal physiological niche for NSC.

4)    The second main criticism, is the fact that authors do not consider all the works published and related to the oligodendrocyte precursor cells or OPC as the cellular origin of glioblastoma stem cells.  Authors have decided to focus on NSC as the cellular origin of BTPC. Anyway, several works  (first paper is Stem Cells, 2011, 29: 590-599) have strong data supporting such a cellular origin. This should be discussed here. 

Author Response

Preamble

We thank the reviewer for the comments and the opportunity to significantly improve our manuscript. In this revision, we have addressed the concerns of the reviewer thoroughly. All changes in the text are indicated in red in the revised manuscript and also depicted in this point to point response. Further, the manuscript was improved by further proofreading.  Please find below our responses to the reviewer’s comments.

Reviewer 2:  

1)        After remembering the clinical situation of GBM patients with their rather limited life expectancy when diagnosis is performed, authors rapidly describe the molecular pathology of GBM and the actual conventional therapy. Then, they move to the NSC and the adult neurogenesis, its possible regulation in various common neurological diseases and its physiological role.  However, although they cite the paper, they do not evoke the possibility that in human brain, this adult neurogenesis is questioned (Sorrells et al., Nature, 2018, 555, 377-381). Indeed, they put this citation in a sentence related to the fact that dentatus gyrus as a neurogenic zone is a still a matter of debate (line 102-104). 

We agree with the reviewer that adult neurogenesis is still a matter of debate in humans. We extended the paragraph about the controversial discussion as following:

“While adult neurogenesis in the SVZ of humans is widely accepted (Sanai, Nguyen et al. 2011, Wang, Liu et al. 2011), it is still under discussion in the hippocampus (Roy, Wang et al. 2000, Sanai, Nguyen et al. 2011, Boldrini, Fulmore et al. 2018, Sorrells, Paredes et al. 2018). Recently, Sorrells et al. (2018) showed that neurogenesis in the dentate gyrus declines sharply in children and that in the adult brain they could not detect any new neurons (Sorrells, Paredes et al. 2018). However, shortly after this publication, Boldrini et al. (2018) published a manuscript stating that neurogenesis still persists throughout life (Boldrini, Fulmore et al. 2018). Thus, it is still controversial if the potential to produce new neurons still exists in the adult brain; but many studies show that it is drastically decreased as compared to embryonic stages (Sanai, Tramontin et al. 2004, Quiñones-Hinojosa, Sanai et al. 2006, Guerrero-Cazares, Gonzalez-Perez et al. 2011, Sanai, Nguyen et al. 2011).”

2)        In Line 241 to 245, authors compare the genes expressed by both NSC and BTPC. They should also mention that these markers could also be expressed by other cell types.  In others words, these are not bona fidemarkers of NSXC and/or BTPC.

We acknowledge that these markers are not uniquely expressed in NSCs and BTPCs, but that rather a pattern of markers may define a subpopulation of cells. We summarized some prominent markers, which can be found in both, NSCs and BTPCs, but it is correct, that individual markers mentioned in the text can be found in various other cell types. Therefore, we changed the text as following:

 “However, these genes are not exclusively expressed in NSCs and BTPCs, but can be found in various other cell types, e.g. GFAP is expressed in astrocytes and radial glial cells.”

3)        The sphere forming capacity is a functional marker of immature cells, normal and cancerous.  The paragraph between line 254 to 265 should be more carefully discussed. Indeed, to sustain the affirmation that glioma arise from NSC based on this property and on a paper published in 2003 is rather weak (line 259-260).

We can understand that our assumption that an involvement of the SVZ automatically indicates an involvement of NSCs was rashly. More precisely, we wanted to indicate rather the impact of the SVZ than of the NSCs themselves. We changed the paragraph in order to clarify the issue as following:

“Following this study, various groups could isolate and culture stem cells from human brain tumor tissue, which have the capacity of tumor regeneration (Ignatova, Kukekov et al. 2002, Hemmati, Nakano et al. 2003, Singh, Clarke et al. 2003, Galli, Binda et al. 2004). Altogether, this led to the assumption that gliomas might arise from stem or progenitor cells located in the SVZ (Recht, Jang et al. 2003, Yip, Sabetrasekh et al. 2006, Quinones-Hinojosa and Chaichana 2007).”  

4)        The first main criticism to this review is the fact that authors do not consider other niches than VZ for glioblastoma stem cells, like vascular or the hypoxic peri-arterioalr niche (several references).  Indeed, it would have been useful considering BTPC and SVZ to compare those glioblastoma-specific niche to SVZ or the normal physiological niche for NSC.

In this review, we want to describe that NSCs from the SVZ could be one possibility as cell of origin for BTPCs in glioblastoma but not that the SVZ itself is the niche for BTPCs. The BTPCs themselves form their own niches within the tumor and different reviews are available for the detailed description of those. Nevertheless, we recognize that these niches should be mentioned and included the following paragraph in the chapter 4.2:

“Those BTPC niches are similar to NSC niches with a specific protective microenvironment composed of specific cell types. Based on their location within the tumor and their composition, they are mostly described as peri-vascular, necrotic and hypoxic niches (Calabrese, Poppleton et al. 2007, Seidel, Garvalov et al. 2010). BTPCs in peri-vascular niches are closely located to endothelial cells and express vascular endothelial growth factor VEGF, a well-known growth factor which regulates angiogenesis (Sharma and Shiras 2016). Those endothelial cells include capillaries, venules, lymph vessels or arterioles, whereby in case of arterioles, the niche can be directly specified as peri-arteriolar niche (Hira, Ploegmakers et al. 2015). In hypoxic niches, the level of hypoxia-inducible factor HIF is increased, which can increase pro-angiogenic growth factors (Li, Bao et al. 2009, Filatova, Seidel et al. 2016). Most recently, a combined concept was presented, comprising all BTPC niches to one integral hypoxic peri-arteriolar niche model (reviewed in (Aderetti, Hira et al. 2018)).

Additionally, we described the physiological niche for NSCs in the SVZ and its cellular composition more in detail (3.2):

“The NSCs in the SVZ are found in an astrocytic ribbon in the sub-ependymal zone. They are surrounded by ependymal cells, vascular endothelial cells, astrocytes and oligodendrocytes (Sanai, Tramontin et al. 2004), which are important to support the stem cells and to control the proliferation rate (Lim and Alvarez-Buylla 1999, Lim, Tramontin et al. 2000). Next to the SVZ, the brain parenchyma is located, which is mostly composed of differentiated neurons and glia cells.

5)        The second main criticism, is the fact that authors do not consider all the works published and related to the oligodendrocyte precursor cells or OPC as the cellular origin of glioblastoma stem cells.  Authors have decided to focus on NSC as the cellular origin of BTPC. Anyway, several works  (first paper is Stem Cells, 2011, 29: 590-599) have strong data supporting such a cellular origin. This should be discussed here.

We agree with the reviewer, that OPCs are quite important and also a likely candidate for the cell of origin in glioblastoma. We rather thought that this review gives an overview over the involvement of NSCs, only, instead of comparing different possibilities for the cell of origin. Describing the involvement of OPCs would fill another review and would go beyond the scope of this article. However, inspired by the reviewer, we included a paragraph (5.2) about other cells of origin in glioblastoma:

“5.2 Committed precursor cells as the cell of origin in glioblastoma

Another theory about the cell of origin states that glioblastoma BTPCs arise from committed precursor cells, like APCs or OPCs. Proliferating multipotent NSCs can create committed neuronal precursor cells that may further divide and differentiate into mature neurons and committed glial precursor cells (GPCs) that can further differentiate into OPCs to generate mature oligodendrocytes and into APCs to generate mature astrocytes (Rowitch and Kriegstein 2010, Jiang and Uhrbom 2012).

For a long time, astrocytes were believed to be the only proliferating cells in the adult brain (Cavanagh 1970) und thus, were subject in glioblastoma research, also because of high GFAP levels in glioma tissue (Jones, Bigner et al. 1981). However, these mature astrocytes would need to dedifferentiate to become tumorigenic, which is possible but unlikely (Sharif, Legendre et al. 2007, Dufour, Cadusseau et al. 2009). Furthermore, GFAP is also expressed by radial glial cells (Alves, Barone et al. 2002) and NSCs of the adult SVZ (Lim and Alvarez-Buylla 1999).

Lindberg et al. (2009) and Hide et al. (2011) introduced OPCs as the cell of origin by specific mouse models to study OPCs in glioblastoma development, like the MADM-based lineage tracing model to mutate sporadically p53/Nf1 (Lindberg, Kastemar et al. 2009, Hide, Takezaki et al. 2011). Hide et al. suggested that transformation of both OPCs and NSCs could lead to formation of BTPCs with tumorigenic properties (Hide, Takezaki et al. 2011). Additionally, Liu et al. noted that it is important to analyze premalignant stages of tumors to identify the cell of origin, because the tumor cell could acquire plasticity and veil their origin. Further, they demonstrated that OPCs, but not NSC or any other NSC-derived lineage, show aberrant growth prior to malignancy. In OPCs are some overlapping marker expressions, like PDGFRα and NG2, which are involved in development of OPCs and are altered in glioma (Shoshan, Nishiyama et al. 1999, 2008, Parsons, Jones et al. 2008). OPCs might form a stem cell niche at the tumor border, increasing chemo-radioresistance and promoting recurrence (Parsons, Jones et al. 2008).

Taken together, the current data suggests that BTPCs might develop from various stem or progenitor cells, which needs to be considered when developing treatment strategies. NSCs, ASCs, OPCs and GPCs might all be the cell of origin, which lead to the development of GB. However, most recent studies using state of the art techniques clearly point to an involvement of NSCs in GB (Lee, Lee et al. 2018).”

(2008). "Comprehensive genomic characterization defines human glioblastoma genes and core pathways." Nature 455(7216): 1061-1068.

Aderetti, D. A., V. V. V. Hira, R. J. Molenaar and C. J. F. van Noorden (2018). "The hypoxic peri-arteriolar glioma stem cell niche, an integrated concept of five types of niches in human glioblastoma." Biochim Biophys Acta Rev Cancer 1869(2): 346-354.

Alves, J. A., P. Barone, S. Engelender, M. M. Froes and J. R. Menezes (2002). "Initial stages of radial glia astrocytic transformation in the early postnatal anterior subventricular zone." J Neurobiol 52(3): 251-265.

Boldrini, M., C. A. Fulmore, A. N. Tartt, L. R. Simeon, I. Pavlova, V. Poposka, G. B. Rosoklija, A. Stankov, V. Arango, A. J. Dwork, R. Hen and J. J. Mann (2018). "Human Hippocampal Neurogenesis Persists throughout Aging." Cell Stem Cell 22(4): 589-599 e585.

Calabrese, C., H. Poppleton, M. Kocak, T. L. Hogg, C. Fuller, B. Hamner, E. Y. Oh, M. W. Gaber, D. Finklestein, M. Allen, A. Frank, I. T. Bayazitov, S. S. Zakharenko, A. Gajjar, A. Davidoff and R. J. Gilbertson (2007). "A perivascular niche for brain tumor stem cells." Cancer Cell 11(1): 69-82.

Cavanagh, J. B. (1970). "The proliferation of astrocytes around a needle wound in the rat brain." J Anat 106(Pt 3): 471-487.

Dufour, C., J. Cadusseau, P. Varlet, A. L. Surena, G. P. de Faria, A. Dias-Morais, N. Auger, N. Leonard, E. Daudigeos, C. Dantas-Barbosa, J. Grill, V. Lazar, P. Dessen, G. Vassal, V. Prevot, A. Sharif, H. Chneiweiss and M. P. Junier (2009). "Astrocytes reverted to a neural progenitor-like state with transforming growth factor alpha are sensitized to cancerous transformation." Stem Cells 27(10): 2373-2382.

Filatova, A., S. Seidel, N. Bogurcu, S. Graf, B. K. Garvalov and T. Acker (2016). "Acidosis Acts through HSP90 in a PHD/VHL-Independent Manner to Promote HIF Function and Stem Cell Maintenance in Glioma." Cancer Res 76(19): 5845-5856.

Galli, R., E. Binda, U. Orfanelli, B. Cipelletti, A. Gritti, S. De Vitis, R. Fiocco, C. Foroni, F. Dimeco and A. Vescovi (2004). "Isolation and characterization of tumorigenic, stem-like neural precursors from human glioblastoma." Cancer Res 64(19): 7011-7021.

Guerrero-Cazares, H., O. Gonzalez-Perez, M. Soriano-Navarro, G. Zamora-Berridi, J. M. Garcia-Verdugo and A. Quinones-Hinojosa (2011). "Cytoarchitecture of the lateral ganglionic eminence and rostral extension of the lateral ventricle in the human fetal brain." J Comp Neurol 519(6): 1165-1180.

Hemmati, H. D., I. Nakano, J. A. Lazareff, M. Masterman-Smith, D. H. Geschwind, M. Bronner-Fraser and H. I. Kornblum (2003). "Cancerous stem cells can arise from pediatric brain tumors." Proceedings of the National Academy of Sciences 100(25): 15178-15183.

Hide, T., T. Takezaki, Y. Nakatani, H. Nakamura, J.-i. Kuratsu and T. Kondo (2011). "Combination of a Ptgs2 Inhibitor and an Epidermal Growth Factor Receptor-Signaling Inhibitor Prevents Tumorigenesis of Oligodendrocyte Lineage-Derived Glioma-Initiating Cells." STEM CELLS 29(4): 590-599.

Hira, V. V., K. J. Ploegmakers, F. Grevers, U. Verbovsek, C. Silvestre-Roig, E. Aronica, W. Tigchelaar, T. L. Turnsek, R. J. Molenaar and C. J. Van Noorden (2015). "CD133+ and Nestin+ Glioma Stem-Like Cells Reside Around CD31+ Arterioles in Niches that Express SDF-1alpha, CXCR4, Osteopontin and Cathepsin K." J Histochem Cytochem 63(7): 481-493.

Ignatova, T. N., V. G. Kukekov, E. D. Laywell, O. N. Suslov, F. D. Vrionis and D. A. Steindler (2002). "Human cortical glial tumors contain neural stem-like cells expressing astroglial and neuronal markers in vitro." Glia 39(3): 193-206.

Jiang, Y. and L. Uhrbom (2012). "On the origin of glioma." Upsala journal of medical sciences 117(2): 113-121.

Jones, T. R., S. H. Bigner, S. C. Schold, Jr., L. F. Eng and D. D. Bigner (1981). "Anaplastic human gliomas grown in athymic mice. Morphology and glial fibrillary acidic protein expression." Am J Pathol 105(3): 316-327.

Lee, J. H., J. E. Lee, J. Y. Kahng, S. H. Kim, J. S. Park, S. J. Yoon, J.-Y. Um, W. K. Kim, J.-K. Lee, J. Park, E. H. Kim, J.-H. Lee, J.-H. Lee, W.-S. Chung, Y. S. Ju, S.-H. Park, J. H. Chang, S.-G. Kang and J. H. Lee (2018). "Human glioblastoma arises from subventricular zone cells with low-level driver mutations." Nature 560(7717): 243-247.

Li, Z., S. Bao, Q. Wu, H. Wang, C. Eyler, S. Sathornsumetee, Q. Shi, Y. Cao, J. Lathia, R. E. McLendon, A. B. Hjelmeland and J. N. Rich (2009). "Hypoxia-inducible factors regulate tumorigenic capacity of glioma stem cells." Cancer Cell 15(6): 501-513.

Lim, D. A. and A. Alvarez-Buylla (1999). "Interaction between astrocytes and adult subventricular zone precursors stimulates neurogenesis." Proc Natl Acad Sci U S A 96(13): 7526-7531.

Lim, D. A., A. D. Tramontin, J. M. Trevejo, D. G. Herrera, J. M. Garcia-Verdugo and A. Alvarez-Buylla (2000). "Noggin antagonizes BMP signaling to create a niche for adult neurogenesis." Neuron 28(3): 713-726.

Lindberg, N., M. Kastemar, T. Olofsson, A. Smits and L. Uhrbom (2009). "Oligodendrocyte progenitor cells can act as cell of origin for experimental glioma." Oncogene 28(23): 2266-2275.

Parsons, D. W., S. Jones, X. Zhang, J. C. Lin, R. J. Leary, P. Angenendt, P. Mankoo, H. Carter, I. M. Siu, G. L. Gallia, A. Olivi, R. McLendon, B. A. Rasheed, S. Keir, T. Nikolskaya, Y. Nikolsky, D. A. Busam, H. Tekleab, L. A. Diaz, Jr., J. Hartigan, D. R. Smith, R. L. Strausberg, S. K. Marie, S. M. Shinjo, H. Yan, G. J. Riggins, D. D. Bigner, R. Karchin, N. Papadopoulos, G. Parmigiani, B. Vogelstein, V. E. Velculescu and K. W. Kinzler (2008). "An integrated genomic analysis of human glioblastoma multiforme." Science 321(5897): 1807-1812.

Quinones-Hinojosa, A. and K. Chaichana (2007). "The human subventricular zone: a source of new cells and a potential source of brain tumors." Exp Neurol 205(2): 313-324.

Quiñones-Hinojosa, A., N. Sanai, M. Soriano-Navarro, O. Gonzalez-Perez, Z. Mirzadeh, S. Gil-Perotin, R. Romero-Rodriguez, M. S. Berger, J. M. Garcia-Verdugo and A. Alvarez-Buylla (2006). "Cellular composition and cytoarchitecture of the adult human subventricular zone: A niche of neural stem cells." Journal of Comparative Neurology 494(3): 415-434.

Recht, L., T. Jang, T. Savarese and N. S. Litofsky (2003). "Neural stem cells and neuro-oncology: quo vadis?" J Cell Biochem 88(1): 11-19.

Rowitch, D. H. and A. R. Kriegstein (2010). "Developmental genetics of vertebrate glial–cell specification." Nature 468: 214.

Roy, N. S., S. Wang, L. Jiang, J. Kang, A. Benraiss, C. Harrison-Restelli, R. A. Fraser, W. T. Couldwell, A. Kawaguchi, H. Okano, M. Nedergaard and S. A. Goldman (2000). "In vitro neurogenesis by progenitor cells isolated from the adult human hippocampus." Nat Med 6(3): 271-277.

Sanai, N., T. Nguyen, R. A. Ihrie, Z. Mirzadeh, H. H. Tsai, M. Wong, N. Gupta, M. S. Berger, E. Huang, J. M. Garcia-Verdugo, D. H. Rowitch and A. Alvarez-Buylla (2011). "Corridors of migrating neurons in the human brain and their decline during infancy." Nature 478(7369): 382-386.

Sanai, N., A. D. Tramontin, A. Quinones-Hinojosa, N. M. Barbaro, N. Gupta, S. Kunwar, M. T. Lawton, M. W. McDermott, A. T. Parsa, J. Manuel-Garcia Verdugo, M. S. Berger and A. Alvarez-Buylla (2004). "Unique astrocyte ribbon in adult human brain contains neural stem cells but lacks chain migration." Nature 427(6976): 740-744.

Seidel, S., B. K. Garvalov, V. Wirta, L. von Stechow, A. Schanzer, K. Meletis, M. Wolter, D. Sommerlad, A. T. Henze, M. Nister, G. Reifenberger, J. Lundeberg, J. Frisen and T. Acker (2010). "A hypoxic niche regulates glioblastoma stem cells through hypoxia inducible factor 2 alpha." Brain 133(Pt 4): 983-995.

Sharif, A., P. Legendre, V. Prevot, C. Allet, L. Romao, J. M. Studler, H. Chneiweiss and M. P. Junier (2007). "Transforming growth factor alpha promotes sequential conversion of mature astrocytes into neural progenitors and stem cells." Oncogene 26(19): 2695-2706.

Sharma, A. and A. Shiras (2016). "Cancer stem cell-vascular endothelial cell interactions in glioblastoma." Biochem Biophys Res Commun 473(3): 688-692.

Shoshan, Y., A. Nishiyama, A. Chang, S. Mork, G. H. Barnett, J. K. Cowell, B. D. Trapp and S. M. Staugaitis (1999). "Expression of oligodendrocyte progenitor cell antigens by gliomas: implications for the histogenesis of brain tumors." Proc Natl Acad Sci U S A 96(18): 10361-10366.

Singh, S. K., I. D. Clarke, M. Terasaki, V. E. Bonn, C. Hawkins, J. Squire and P. B. Dirks (2003). "Identification of a cancer stem cell in human brain tumors." Cancer Res 63(18): 5821-5828.

Sorrells, S. F., M. F. Paredes, A. Cebrian-Silla, K. Sandoval, D. Qi, K. W. Kelley, D. James, S. Mayer, J. Chang, K. I. Auguste, E. F. Chang, A. J. Gutierrez, A. R. Kriegstein, G. W. Mathern, M. C. Oldham, E. J. Huang, J. M. Garcia-Verdugo, Z. Yang and A. Alvarez-Buylla (2018). "Human hippocampal neurogenesis drops sharply in children to undetectable levels in adults." Nature 555(7696): 377-381.

Wang, C., F. Liu, Y. Y. Liu, C. H. Zhao, Y. You, L. Wang, J. Zhang, B. Wei, T. Ma, Q. Zhang, Y. Zhang, R. Chen, H. Song and Z. Yang (2011). "Identification and characterization of neuroblasts in the subventricular zone and rostral migratory stream of the adult human brain." Cell Res 21(11): 1534-1550.

Yip, S., R. Sabetrasekh, R. L. Sidman and E. Y. Snyder (2006). "Neural stem cells as novel cancer therapeutic vehicles." Eur J Cancer 42(9): 1298-1308.

Reviewer 3 Report

The review of C. Altmann et al. represents an interesting and comprehensive overview of the role of SVZ in the origin and progression of glioblastomas. This manuscript is a useful contribution in the field for a large audience ranging from basic neuroscientists to clinicians. However, before publication, I suggest that some aspects  need some ameliorations:

Pag.2 lane 46: the authors have written: “Gliomas are divers brain tumors” is the word “divers” a typo? If this is not a typo, please re-formulate the sentence as it is of difficult interpretation.

Pag2 lane 68: instead of “sufficient therapies” it would be better to use “effective therapies”

Pag4 lane 117: “The NSCs in the SVZ are found in an astrocytic ribbon in the sub-ependymal zone. They are surrounded by ependymal cells, vascular endothelial cells, astrocytes, neurons and oligodendrocytes”. Please specify that neurons reside outside the SVZ, in the parenchima, as from the text reported is not clear.

Pag4 lane 138: possibly the word “there” is missing

Pag.4 lane 38 and following paragraphs: the authors mentioned here that there are anatomical differences in the human SVZ compared to rodents. They describe in details the cellular and morphological composition of the human SVZ, but they did not describe or mention the mouse counterpart. What is the main structural/morphological difference of the human SVZ compared with the rodent? For instance is it the number of layer in the murine SVZ different compared to humans?

Pag.6 last paragraph: “As discussed above, stem cells are scarce in the adult brain and can only grow in protective stem cells niches, including the hippocampus and the SVZ [32,33]. These neural precursor cells already possess the ability to proliferate and thus they could transform more easily and rapidly into BTPCs than any other post-mitotic neural cell in the brain [94] “.The term “neural precursor” used here to refer to NSCs is a bit confusing as this is commonly used to collectively describe the mixed population of NSCs and neural progenitor cells.  The authors need to specify if they are referring to just NSCs or the mixed population NSCs and NPCs.

Fig2: the colors used to identify the cancer stem cells and cancer cells are too similar, and then it is difficult to separate these 2 population of cells.

Pag7 lane 215: It has been reported: “As other stem cells, NSCs have the ability of self-renewal and by asymmetric cell division progenitor and differentiated daughter cells are created.” This sentence is confusing: in a typical outcome of an asymmetric division, the stem or progenitor cell generates a copy of itself, which retains self-renewal ability and differentiation potential, and one daughter that enters the path of differentiation. Please correct.

Pag9 lane 284: “Furthermore, the role of chronic inflammation in the SVZ could also contribute to easier mutations of the corresponding NSCs into BTPCs (reviewed in [94]).” This sentence is confusing and needs to be re-formulated.

Pag9 lane 294: please substitute or delete the word “corresponding” at the beginning of the sentence

Pag11 lane 364: please substitute “Notch Signaling” to “Notch signaling”.

Pag11 lane 373:  “the cells’ growth” please correct English

Author Response

Preamble

We thank the reviewer for the comments and the opportunity to significantly improve our manuscript. In this revision, we have addressed the concerns of the reviewer thoroughly. All changes in the text are indicated in red in the revised manuscript and also depicted in this point to point response. Further, the manuscript was improved by further proofreading.  Please find below our responses to the reviewer’s comments.

Reviewer 3:

The review of C. Altmann et al. represents an interesting and comprehensive overview of the role of SVZ in the origin and progression of glioblastomas. This manuscript is a useful contribution in the field for a large audience ranging from basic neuroscientists to clinicians. However, before publication, I suggest that some aspects need some ameliorations:

1.      Pag.2 lane 46: the authors have written: “Gliomas are divers brain tumors” is the word “divers” a typo? If this is not a typo, please re-formulate the sentence as it is of difficult interpretation.

We acknowledge that this sentence was misleading and rephrased it as following:

“Glioma is an umbrella term, compromising around 30 percent of all brain tumors that are thought to grow from intrinsic glia cells.”

2.      Pag2 lane 68: instead of “sufficient therapies” it would be better to use “effective therapies”

In agreement, we changed the terms.

3.      Pag4 lane 117: “The NSCs in the SVZ are found in an astrocytic ribbon in the sub-ependymal zone. They are surrounded by ependymal cells, vascular endothelial cells, astrocytes, neurons and oligodendrocytes”. Please specify that neurons reside outside the SVZ, in the parenchima, as from the text reported is not clear.

We recognize that this cloud be misleading. We rephrased the sentence as following:

“The NSCs in the SVZ are found in an astrocytic ribbon in the sub-ependymal zone. They are surrounded by ependymal cells, vascular endothelial cells, astrocytes and oligodendrocytes (Sanai, Tramontin et al. 2004), which are important to support the stem cells and to control the proliferation rate (Lim and Alvarez-Buylla 1999, Lim, Tramontin et al. 2000). Next to the SVZ, the brain parenchyma is located, which is mostly composed of differentiated neurons and glia cells.”

4.      Pag4 lane 138: possibly the word “there” is missing

In agreement, we added the missing word.

5.      Pag.4 lane 138 and following paragraphs: the authors mentioned here that there are anatomical differences in the human SVZ compared to rodents. They describe in details the cellular and morphological composition of the human SVZ, but they did not describe or mention the mouse counterpart. What is the main structural/morphological difference of the human SVZ compared with the rodent? For instance is it the number of layer in the murine SVZ different compared to humans?

We agree with the author that the cellular and morphological composition of the rodent SVZ was missing, which we included in the text as following:

“In adult rodents, the SVZ contains four major cell types: ependymal cells, NSCs, fast proliferation precursors and neuroblasts. The ependymal cells form a monolayer-border to the ventricle. This layer is followed by the other three cell types, which are not arranged in layers and keep close contact to the ependymal layer. Astrocyte-like NSCs have an apical cilium, which extends into the ventricle lumen and might influence cell proliferation and differentiation (Mirzadeh, Merkle et al. 2008, Ihrie and Álvarez-Buylla 2011, Tong, Han et al. 2014). These astrocyte-like NSCs (type-B cells) occasionally give rise to multipotent intermediate progenitors (type-C cells), which correspond to transit-amplifying cells that further divide to generate neuroblasts (type-A cells) (reviewed in detail in (Fuentealba, Obernier et al. 2012)).”

Furthermore, we highlighted differences between the rodent and the human SVZ in the bulleted list on page 5.

6.      Pag.6 last paragraph: “As discussed above, stem cells are scarce in the adult brain and can only grow in protective stem cells niches, including the hippocampus and the SVZ [32,33]. These neural precursor cells already possess the ability to proliferate and thus they could transform more easily and rapidly into BTPCs than any other post-mitotic neural cell in the brain [94] “.The term “neural precursor” used here to refer to NSCs is a bit confusing as this is commonly used to collectively describe the mixed population of NSCs and neural progenitor cells.  The authors need to specify if they are referring to just NSCs or the mixed population NSCs and NPCs.

We thank the reviewer for the precise reading and changed the phrase neural precursor cell into NSCs. 

7.      Fig2: the colors used to identify the cancer stem cells and cancer cells are too similar, and then it is difficult to separate these 2 population of cells.

We recognize the remark of the reviewer and changed the color of the cancer stem cells in figure 2 and figure 3.

8.      Pag7 lane 215: It has been reported: “As other stem cells, NSCs have the ability of self-renewal and by asymmetric cell division progenitor and differentiated daughter cells are created.” This sentence is confusing: in a typical outcome of an asymmetric division, the stem or progenitor cell generates a copy of itself, which retains self-renewal ability and differentiation potential, and one daughter that enters the path of differentiation. Please correct.

We acknowledge the suggestion and changed “progenitor” into “another NSC”.

9.      Pag9 lane 284: “Furthermore, the role of chronic inflammation in the SVZ could also contribute to easier mutations of the corresponding NSCs into BTPCs (reviewed in [94]).” This sentence is confusing and needs to be re-formulated.

In agreement, we rephrased the sentence as following:

“Furthermore, chronic inflammation in the SVZ could facilitate and accelerate mutations in NSCs and thus contribute to the transformation into BTPCs (reviewed in (Bardella, Al-Shammari et al. 2018)).”  

10.  Pag9 lane 294: please substitute or delete the word “corresponding” at the beginning of the sentence

We deleted the word “corresponding”.

11.  Pag11 lane 364: please substitute “Notch Signaling” to “Notch signaling”.

In agreement, we substituted “Notch Signaling” with “Notch signaling”.

12.  Pag11 lane 373:  “the cells’ growth” please correct English

As suggested, we corrected the expression.

Bardella, C., A. R. Al-Shammari, L. Soares, I. Tomlinson, E. O'Neill and F. G. Szele (2018). "The role of inflammation in subventricular zone cancer." Prog Neurobiol 170: 37-52.

Fuentealba, L. C., K. Obernier and A. Alvarez-Buylla (2012). "Adult neural stem cells bridge their niche." Cell Stem Cell 10(6): 698-708.

Ihrie, Rebecca A. and A. Álvarez-Buylla (2011). "Lake-Front Property: A Unique Germinal Niche by the Lateral Ventricles of the Adult Brain." Neuron 70(4): 674-686.

Lim, D. A. and A. Alvarez-Buylla (1999). "Interaction between astrocytes and adult subventricular zone precursors stimulates neurogenesis." Proc Natl Acad Sci U S A 96(13): 7526-7531.

Lim, D. A., A. D. Tramontin, J. M. Trevejo, D. G. Herrera, J. M. Garcia-Verdugo and A. Alvarez-Buylla (2000). "Noggin antagonizes BMP signaling to create a niche for adult neurogenesis." Neuron 28(3): 713-726.

Mirzadeh, Z., F. T. Merkle, M. Soriano-Navarro, J. M. Garcia-Verdugo and A. Alvarez-Buylla (2008). "Neural Stem Cells Confer Unique Pinwheel Architecture to the Ventricular Surface in Neurogenic Regions of the Adult Brain." Cell Stem Cell 3(3): 265-278.

Sanai, N., A. D. Tramontin, A. Quinones-Hinojosa, N. M. Barbaro, N. Gupta, S. Kunwar, M. T. Lawton, M. W. McDermott, A. T. Parsa, J. Manuel-Garcia Verdugo, M. S. Berger and A. Alvarez-Buylla (2004). "Unique astrocyte ribbon in adult human brain contains neural stem cells but lacks chain migration." Nature 427(6976): 740-744.

Tong, C. K., Y.-G. Han, J. K. Shah, K. Obernier, C. D. Guinto and A. Alvarez-Buylla (2014). "Primary cilia are required in a unique subpopulation of neural progenitors." Proceedings of the National Academy of Sciences 111(34): 12438-12443.

Round 2

Reviewer 2 Report

All my concerns about the first version of the manuscript have been correctly addressed and answered.